# Spin-current probe for phase transition in an insulator

Zhiyong Qiu[1,2], Jia Li[3], Dazhi Hou[1,2], Elke Arenholz[4], Alpha T. N'Diaye[4], Ali Tan[3], Ken-ichi Uchida[5,6], Koji Sato[1], Satoshi Okamoto[7], Yaroslav Tserkovnyak[8], Z.Q. Qiu[3] & Eiji Saitoh[1,2,5,9]

Spin fluctuation and transition have always been one of the central topics of magnetism and condensed matter science. Experimentally, the spin fluctuation is found transcribed onto scattering intensity in the neutron-scattering process, which is represented by dynamical magnetic susceptibility and maximized at phase transitions. Importantly, a neutron carries spin without electric charge, and therefore it can bring spin into a sample without being disturbed by electric energy. However, large facilities such as a nuclear reactor are necessary. Here we show that spin pumping, frequently used in nanoscale spintronic devices, provides a desktop microprobe for spin transition; spin current is a flux of spin without an electric charge and its transport reflects spin excitation. We demonstrate detection of antiferromagnetic transition in ultra-thin CoO films via frequency-dependent spin-current transmission measurements, which provides a versatile probe for phase transition in an electric manner in minute devices.

[1] WPI Advanced Institute for Materials Research, Tohoku University, Sendai 980-8577, Japan. [2] Spin Quantum Rectification Project, ERATO, Japan Science and Technology Agency, Sendai 980-8577, Japan. [3] Department of Physics, University of California at Berkeley, Berkeley, California 94720, USA. [4] Advanced Light Source, Lawrence Berkeley National Laboratory, Berkeley, California 94720, USA. [5] Institute for Materials Research, Tohoku University, Sendai 980-8577, Japan. [6] PRESTO, Japan Science and Technology Agency, Saitama 332-0012, Japan. [7] Materials Science and Technology Division, Oak Ridge National Laboratory, Oak Ridge, Tennessee 37831, USA. [8] Department of Physics and Astronomy, University of California, Los Angeles, California 90095, USA. [9] Advanced Science Research Center, Japan Atomic Energy Agency, Tokai 319-1195, Japan. Correspondence and requests for materials should be addressed to D.H. (email: dazhi.hou@imr.tohoku.ac.jp).

A spin current refers to a flow of spin angular momentum of electrons in condensed matter[1]. There are types of spin currents, including a spin current carried by conduction electrons and one carried by spin waves[1]. The former type of spin current, conduction-electron spin current, can be detected by using the inverse spin Hall effect (ISHE)[2–7], the conversion of a spin current into electric voltage via the spin–orbit interaction in a conductor, typically in Pt. Although conduction-electron spin current can reside only in metals and semiconductors, the latter type of spin current, called spin-wave spin current, can exist even in insulators.[4,8] In fact, spin-wave spin currents have been studied in magnetic insulators and, very recently, in antiferromagnetic alloys and insulators[9–15].

For spin-current generation, one of the most versatile and powerful methods is spin pumping, an induction of spin current from magnetization precession in a magnetic metal into an attached metal via the exchange interaction at the interface[5]. Spin pumping was found to drive spin current also from magnetic insulators, such as $Y_3Fe_5O_{12}$ (refs 4,6,16–22).

Various powerful theories have been constructed to describe the spin-pumping phenomenon[23–25]. They commonly predict that the efficiency of spin pumping is sensitive to dynamical magnetic susceptibility at the interface between a magnet and a metal in a spin-pumping system. Therefore, spin pumping is sensitive to interface magnetic susceptibility, while standard magnetometry probes bulk properties which often hides interface signals. This raises an interesting hypothesis: when a very thin sample film is inserted at the interface of a spin-pumping system, spin pumping may reflect the dynamical susceptibility, which is directly related to spin fluctuation according to the fluctuation-dissipation theory, of the inserted thin sample film. Here we show that this is the case by using an antiferromagnetic transition in an ultra-thin film of CoO, and that spin pumping becomes an *in situ* microprobe for magnetic phase transition.

## Results

**Sample description.** Figure 1c is a schematic illustration of the sample system used in the present study; we inserted an antiferromagnetic CoO thin film at the interface between $Y_3Fe_5O_{12}$ and Pt layers in a typical spin-pumping system $Y_3Fe_5O_{12}/Pt$ to form $Y_3Fe_5O_{12}/CoO/Pt$. At low temperatures, CoO exhibits antiferromagnetic order[26]. Here, $Y_3Fe_5O_{12}$ is a

typical spin-pumping material, by which spin current is emitted when magnetization precession is excited[4,6]. Pt is used as a spin-current detector based on ISHE, in which a spin current is converted into an electric voltage in Pt perpendicular to the spin-current spin polarization direction[2–4,6]. When magnetization precession in $Y_3Fe_5O_{12}$ is excited by a microwave application, spin pumping is driven and then a spin current is injected from the $Y_3Fe_5O_{12}$ layer into the Pt layer across the thin CoO layer[23–25].

**Spin-pumping signal of $Y_3Fe_5O_{12}/CoO/Pt$ system.** Figure 2a shows the magnetic field dependence of microwave absorption spectra of $Y_3Fe_5O_{12}$ at various temperatures measured when a 5 GHz microwave is applied. At $T = 300$ K, absorption peaks appear around $H_{FMR} = \pm 1.2$ KOe, which correspond to ferromagnetic resonance (FMR) in the $Y_3Fe_5O_{12}$. With decreasing temperature, $H_{FMR}$ is observed to slightly decrease, which is due to the temperature dependence of magnetization in the $Y_3Fe_5O_{12}$. The microwave absorption power $P_{ab}$ at $H_{FMR}$ is almost constant with changing $T$.

In Fig. 2b, we show the voltage $V$ generated in the Pt layer in a simple $Y_3Fe_5O_{12}/Pt$ spin-pumping system without a CoO layer measured by applying a 5 GHz microwave. At the FMR field $H_{FMR}$, a clear voltage peak appears at all temperatures. The sign of the peak voltage $V_{ISHE}$ is reversed by reversing the polarity of the applied magnetic field, showing that the voltage peak is due to ISHE induced by spin current pumped from the $Y_3Fe_5O_{12}$ layer[2–4,23–25].

Figure 2d shows the temperature dependence of the peak voltage $V_{ISHE}$ for the $Y_3Fe_5O_{12}/Pt$ film without a CoO layer. $V_{ISHE}$ decreases monotonically with decreasing the temperature. This monotonic decrease is attributed to the decrease in the resistivity of Pt and the increase in the magnetization damping in $Y_3Fe_5O_{12}$.

On the contrary by inserting a CoO layer in the simple spin-pumping system, a clear unconventional peak structure appears in the temperature dependence of the ISHE signal (Fig. 2c). Figure 2e shows the temperature dependence of the voltage peak intensity $V_{ISHE}$ for the $Y_3Fe_5O_{12}/CoO/Pt$ trilayer film. The $V_{ISHE}$ for the $Y_3Fe_5O_{12}/CoO/Pt$ trilayer film exhibits a clear peak at $T = 200$ K, which is quite different from that for the $Y_3Fe_5O_{12}/Pt$ bilayer film (Fig. 2d). The $V_{ISHE}$ peak temperature is comparable to the Néel temperature of the CoO layer determined by an X-ray magnetic linear dichroism

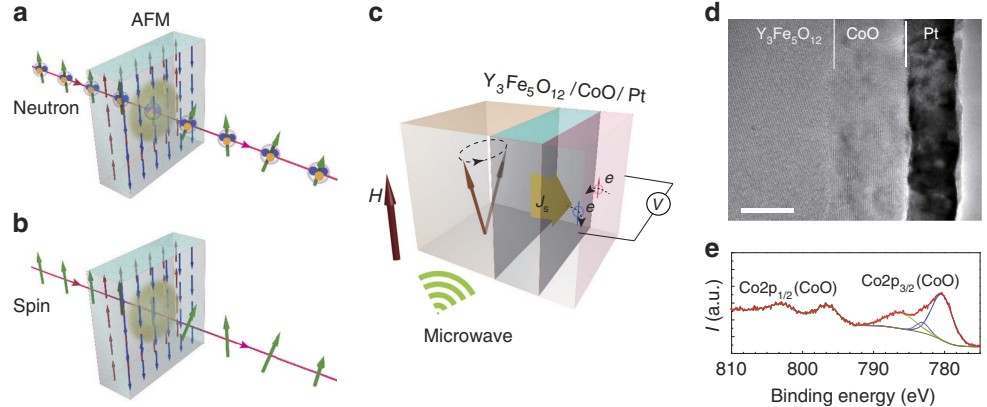

**Figure 1 | Concept and sample set-up.** (**a**) Inelastic scattering of polarized neutrons through an antiferromagnetic system. (**b**) Spin-current transmission through an antiferromagnetic system. (**c**) Experimental set-up of the spin-pumping measurement for the $Y_3Fe_5O_{12}/CoO/Pt$ trilayer device. $J_s$ denotes spin current injected from the $Y_3Fe_5O_{12}$ layer into the Pt layer through the CoO layer by spin pumping, which is detected as a voltage signal via the inverse spin Hall effect in the Pt layer. (**d**) A cross-sectional TEM image of a $Y_3Fe_5O_{12}/CoO/Pt$ trilayer device. Scale bar, 10 nm. (**e**). A Co 2p XPS spectrum and Gaussian fitting analysis for the CoO layer in the $Y_3Fe_5O_{12}/CoO/Pt$ trilayer device. TEM, transmission electron microscopy; XPS, X-ray photoemission spectroscopy.

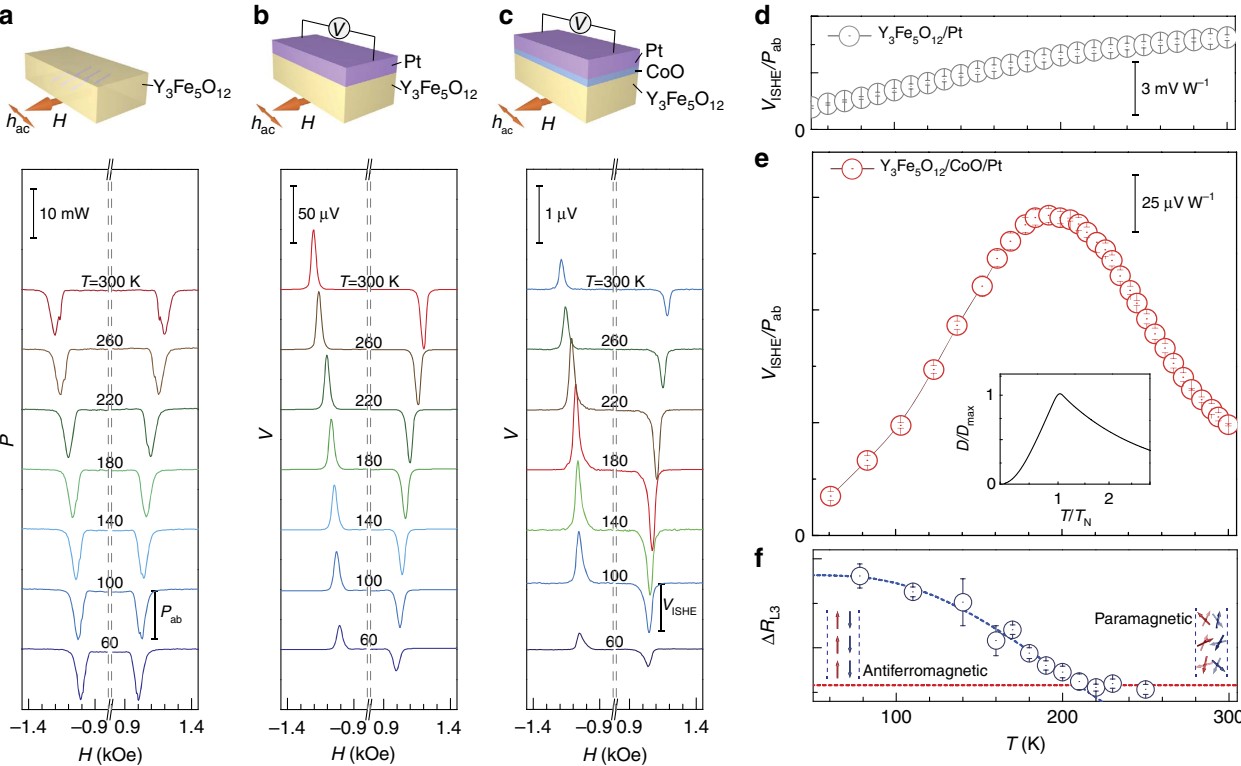

**Figure 2 | Spin-pumping detection of antiferromagnetic transition.** (**a**) Magnetic field ($H$) dependence of microwave absorption power ($P$) for a $Y_3Fe_5O_{12}$ film (3 μm in thickness) at various temperatures. $P_{ab}$ denotes absorption power at FMR field. (**b**) Magnetic field ($H$) dependence of electric voltage ($V$) generated in the $Y_3Fe_5O_{12}$ (3 μm)/Pt (10 nm) bilayer film at various temperatures. (**c**) Magnetic field ($H$) dependence of electric voltage ($V$) generated in the $Y_3Fe_5O_{12}$ (3 μm)/CoO (6 nm)/Pt (10 nm) trilayer film at various temperatures. $V_{ISHE}$ denotes the voltage signal at the FMR field. (**d**) Temperature dependence of $V_{ISHE}$ for the $Y_3Fe_5O_{12}$ (6 μm)/Pt (10 nm) bilayer film. (**e**) Temperature dependence of $V_{ISHE}$ for the $Y_3Fe_5O_{12}$ (3 μm)/CoO (6 nm)/Pt (10 nm) trilayer film. The inset shows the theoretical prediction of the spin conductance versus temperature in an antiferromagnetic system with $S = 1/2$ (ref. 38). Coefficient $D$ denotes the spin conductance at a given frequency scaled by its maximum value $D_{max}$. (**f**) Temperature dependence of the XMLD signal $\Delta R_{L3}$ for the $Y_3Fe_5O_{12}$ (3 μm)/CoO (6 nm)/Pt (1 nm) trilayer film (details are shown in Supplementary Note 1). The error bars in **d**-**f** represent the s.d. of multiple measurements at the same condition. XMLD, X-ray magnetic linear dichroism.

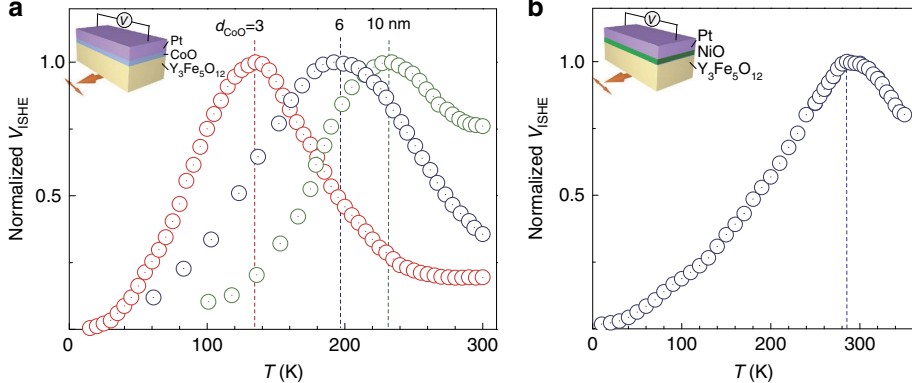

**Figure 3 | Temperature dependence of spin-pumping signals in different systems.** (**a**) Temperature dependence of $V_{ISHE}$ for $Y_3Fe_5O_{12}$/CoO/Pt trilayer films with different CoO-layer thicknesses ($d_{CoO} = 3$, 6 and 10 nm). (**b**) Temperature dependence of $V_{ISHE}$ for a $Y_3Fe_5O_{12}$/NiO(1.5 nm)/Pt film. The dash lines in **a**,**b** denote the peak positions.

measurement using a synchrotron facility as shown in Fig. 2f (for detailed data, please see Supplementary Note 1 and Supplementary Figs 1 and 2). Furthermore, the temperature dependence of $V_{ISHE}$ is similar to that of the magnetic susceptibility in a bulk CoO, in which the susceptibility is maximized around the Néel temperature $T_N$ (refs 27,28).

**Spin pumping with different CoO layer thickness.** Figure 3a shows the temperature dependence of $V_{ISHE}$ measured for various thicknesses of the CoO layer in $Y_3Fe_5O_{12}$/CoO/Pt trilayer films. In CoO films, the Néel temperature is known to decrease with decreasing the thickness of the film due to the finite size effect[29,30]. The observed CoO-thickness dependence of the peak

temperature is consistent with this feature: the peak temperature decreases with decreasing the CoO layer thickness. All the results show that the $V_{ISHE}$ peak position indicates the Néel temperature of the CoO layer, and that the $V_{ISHE}$ enhancement around the antiferromagnetic transition can be related to the CoO-film susceptibility enhancement, which is a good measure for spin fluctuations.

**Spin-pumping signal of $Y_3Fe_5O_{12}$/NiO/Pt system.** To check the universality of the phenomenon, we measured another antiferromagnet: a NiO film (1.5 nm) in a $Y_3Fe_5O_{12}$/NiO/Pt system and found a similar peak structure and temperature dependence of $V_{ISHE}$ (Fig. 3b). The peak position is consistent with the previous study on the Néel temperature in ultra-thin NiO films[31]. Furthermore, when a Cu layer is inserted between the $Y_3Fe_5O_{12}$ and the CoO (NiO) layers, we observed similar peak structures in the $T$-dependent ISHE signal (Supplementary Note 2 and Supplementary Fig. 3). This result suggests that the direct exchange coupling between $Y_3Fe_5O_{12}$ and an antiferromagnet is not necessary for the $V_{ISHE}$ enhancement around Néel temperatures. Although observing such a phase transition in a single ultra-thin film was impossible without using large synchrotron facilities and a special X-ray magnetic linear dichroism spectrometer[32–34], our present method provides a way to probe it by a table-top experiment.

## Discussion

Our present study stimulates further investigations of not only the spin transport near magnetic phase transitions but also microscopic spin-transport properties in antiferromagnetic systems. In antiferromagnetic insulators, incoherent thermal magnons and coherent Néel-order parameter dynamics[12] are considered to be responsible for transporting spins. Our experimental results in Fig. 3, however, show that $V_{ISHE}$ is strongly suppressed towards lower temperatures in both cases of $Y_3Fe_5O_{12}$/CoO/Pt and $Y_3Fe_5O_{12}$/NiO/Pt systems; $V_{ISHE}$ at 10 K is much less than that at $T_N$. Also, we notice that this feature was confirmed by some recent studies[35,36]. These results indicate that the spins are transported dominantly by incoherent thermal magnons rather than coherent Néel dynamics. At high temperatures, thermal magnons continuously evolve into thermal spin fluctuations, which would transport spin current above Néel temperature.

Such thermal spin dynamics both below and above $T_N$ are well described by a bosonic auxiliary particle method[37]. Using this method, the spin conductivity in an antiferromagnetic insulator was shown to be maximized near its Neel temperature[38], exactly like our $V_{ISHE}$ (Fig. 2e inset). Since $V_{ISHE}$ measures spin moments transferred across magnetic insulators, its enhancement directly reflects that of the spin conductivity. The spin conductivity and the magnetic susceptibility are in principle different quantities. However, their temperature dependences are rather similar because both are dominated by spin excitations with zero momentum transfer. Therefore, $V_{ISHE}$ in our experimental set-up is a good measure for the spin dynamics and transition.

Moreover, another significant remark on our result which requires further theoretical understanding is the frequency dependence of the spin-pumping behaviour. Figure 4a shows the microwave frequency $f$ as well as the temperature

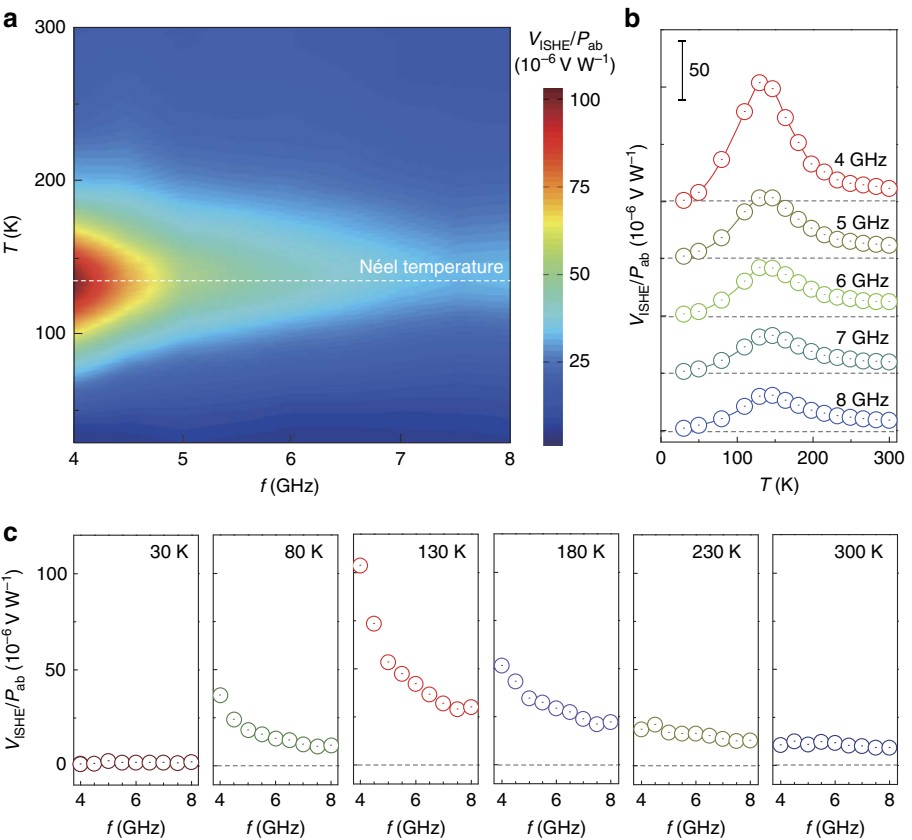

**Figure 4 | Frequency dependence of spin-pumping signals in $Y_3Fe_5O_{12}$/CoO/Pt.** (**a**) A pseudo-colour plot of $V_{ISHE}/P_{ab}$ as a function of the temperature $T$ and the microwave frequency $f$ for the $Y_3Fe_5O_{12}$ (3 μm)/CoO (3 nm)/Pt (10 nm) trilayer film. (**b**) Temperature ($T$) dependence of $V_{ISHE}/P_{ab}$ at various exciting microwave frequencies. (**c**) Exciting microwave frequency ($f$) dependence of $V_{ISHE}/P_{ab}$ at various temperatures. We did not show the data for frequencies lower than 4 GHz because the magnetization precession can be modulated due to the three-magnon interaction when $f < 4$ GHz (ref. 40).

$T$ dependence of $V_{ISHE}/P_{ab}$ for the $Y_3Fe_5O_{12}/CoO/Pt$ trilayer film, where $V_{ISHE}$ is normalized by the FMR microwave absorption $P_{ab}$. For all the frequencies, $V_{ISHE}/P_{ab}$ shows peaks around the Néel temperature $T_N \sim 130$ K. As seen in Fig. 4c, $V_{ISHE}/P_{ab}$ exhibits different $f$ dependence at different temperatures, namely $V_{ISHE}/P_{ab}$ strongly depends on $f$ only near $T_N \sim 130$ K, but it weakly depends on $f$ at temperatures far from $T_N$. Such a strong frequency dependence implies that the observed phenomena reflect dynamical properties. Similar frequency dependence was also observed in the $Y_3Fe_5O_{12}/NiO/Pt$ trilayer film, but it was absent in the $Y_3Fe_5O_{12}/Pt$ bilayer film without antiferromagnetic layers (Supplementary Note 3 and Supplementary Fig. 4), showing that the observed prominent frequency dependence is characteristic to the antiferromagnetic layers near $T_N$. This type of strong frequency dependence cannot be explained by the coherent Néel dynamics or simple thermal magnons alone. First, coherent precession of $Y_3Fe_5O_{12}$ can pump antiferromagnetic magnons at the $Y_3Fe_5O_{12}$/antiferromagnetic interface, in analogy to the conventional spin pumping at the ferromagnet/metal interfaces[12]. Second, the FMR dynamics inside of $Y_3Fe_5O_{12}$ itself should pump thermal magnons, which can subsequently diffuse across the structure, inducing the ISHE signal. This latter process relies on the breaking of the SU(2) symmetry of magnetic dynamics[39], which would be progressively enhanced with the increased ellipticity of the coherent dynamics in $Y_3Fe_5O_{12}$. Increased ellipticity of the Kittel mode in $Y_3Fe_5O_{12}$ at lower frequencies can thus be responsible for the enhancement of the conversion of the coherent precession into thermal magnons, leading to the observed increase in the signal. Alternatively, the finite lifetime of thermal magnons could be a source of the observed frequency dependence as in the case of the frequency-dependent spin conductance[38]. In principle, the magnon lifetime depends on temperature and other intrinsic and extrinsic effects. A detailed theoretical analysis of the interplay of the coherent and incoherent magnetic dynamics in our heterostructure is, however, beyond the scope of this work. Constructing a comprehensive theory for spin-current transport in magnetic heterostructures is an important outstanding task for the development of novel spintronics based on quantum magnets.

## Methods

**Preparation of $Y_3Fe_5O_{12}/CoO/Pt$ samples.** A 3 µm-thick single-crystalline $Y_3Fe_5O_{12}$ film was grown on a (111) $Gd_3Ga_5O_{12}$ wafer by a liquid phase epitaxy method at 1,203 K in a $PbO-B_2O_3$ based flux. All the samples were cut from a same wafer into $1.5 \times 3$ mm$^2$ in size. CoO films with differencit thicknesses were coated on the $Y_3Fe_5O_{12}$ film by a radio frequency magnetron sputtering method. All the CoO films were prepared at 1,073 K to restrain the formation of cobalt oxide with other valence states and to improve the crystallinity. Then, 10-nm-thick Pt films were put on the top of the CoO films with a Hall-bar structure by an radio frequency magnetron sputtering method.

**Sample characterization.** Crystallographic characterization for samples was carried out by a X-ray diffractometry and transmission electron microscopy. A transmission electron microscopy image shows that the $Y_3Fe_5O_{12}$ film is of a single-crystal structure, and CoO and Pt layers are nearly epitaxially grown on the $Y_3Fe_5O_{12}$ film (Fig. 1d). An X-ray photoelectron spectroscopy method was used to confirm the chemical valence of the CoO layer (Fig. 1e).

**Spin-pump experimental set-up.** The spin-pumping measurement was performed in a physical property measurement system (PPMS), Quantum Design, Inc. To excite FMR in the $Y_3Fe_5O_{12}$ layer, microwave was applied by using a coplanar waveguide. The voltage signal between the ends of the Pt layer was measured by using a lock-in amplifier. Temperature dependence measurement was carried out from 10 to 300 K, after cooling samples from room temperature to 10 K in a 5,000 Oe magnetic field.

**Data availability.** All relevant data are available from the corresponding author on request.

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

## Acknowledgements

This work was supported by JST-ERATO 'Spin Quantum Rectification', JST-PRESTO 'Phase Interfaces for Highly Efficient Energy Utilization', Grant-in-Aid for Scientific Research on Innovative Area, 'Nano Spin Conversion Science' (26103005 and 26103006), Grant-in-Aid for Scientific Research (S) (25220910), Grant-in-Aid for Scientific Research (A) (25247056 and 15H02012), Grant-in-Aid for Challenging Exploratory Research (26600067), Grant-in-Aid for Research Activity Start-up (25889003), and World Premier International Research Center Initiative (WPI), all from MEXT, Japan, the ImPACT program of the Council for Science, Technology and Innovation, Cabinet Office, Japan, and NEC corporation. Financial support from National Science Foundation DMR-1504568, Future Materials Discovery Program through the National Research Foundation of Korea (No. 2015M3D1A1070467), and Science Research Center Program through the National Research Foundation of Korea (No. 2015R1A5A1009962) is gratefully acknowledged (J. L., A. T., Z. Q.). The Advanced Light Source is supported by the US Department of Energy under contract number DE-AC02-05CH11231 (E.A., A.N.). The research by S.O. is supported by the US Department of Energy, Office of Science, Basic Energy Sciences, Materials Sciences and Engineering Division. Y.T is supported by U.S. Department of Energy, Office of Basic Energy Sciences under Award No. DE-SC0012190.

## Author contributions

Z.Q. and D.H. designed the experiment, fabricated the samples, collected all of the data and analysed the data. E.S. supervised this study. J.L., Z.Q.Q., E.A., A.T.N.D. and A.T. performed the XMLD and XMCD measurements. K.S., S.O. and Y.T. contribute theoretical suggestions. K.U. helped the low-temperature experiment. Z.Q., D.H., E.S., S.O., K.S., K.U. and Y.T. wrote the manuscript. All the authors discussed the results and commented on the manuscript.

## Additional information

**Competing financial interests:** The authors declare no competing financial interests.

