## [Peer Review File · Nature Communications]

REVIEWERS' COMMENTS:

Reviewer #1 (Remarks to the Author):

This manuscript is now well revised by taking into account the suggestions and comments given by referees in the first round of reviews for [redacted]. So I believe that the manuscript warrant publication in Nature Communications.

Reviewer #2 (Remarks to the Author):

The authors have revised their manuscript according to the input of reviewers. Now the comparison with Neutron scattering is made in a more appropriate way. As I stated in my previous report, many of the authors of this work are known experts in the field and this is a very solid study worthy publishing. However, with the delay of re-submission to another journal, the novelty of this work is further compromised. While I was re-evaluating this work for Nature Communications, it has come to my attention that a very similar study on the temperature dependence has been published in Physical Review Letters last month: PRL 116, 186601 (2016). The very similar conclusion with the present manuscript was drawn in the PRL paper. Therefore I couldn't recommend publication of this work in Nature Communications.

Reviewer #3 (Remarks to the Author):

The authors have improved the manuscript by incorporating a connection to the possible mechanisms (which were asserted before and not explicitly explained).

Although the paper remains primarily experimental with many remaining questions, its results and the connection of the AFM fluctuations to the enhancement of the ISHE is well established by the data and the systematic methodology of the experiments (including the Cu inserted sample).

I am satisfied with the reply to my previous comments and I am happy to recommend the the article for publication in nature Communications.

Comments and responses (Second Run)

Referee 1:

Comment:

This manuscript is now well revised by taking into account the suggestions and comments given by referees in the first round of reviews for [redacted]. So I believe that the manuscript warrant publication in Nature Communications.

Ans: Thanks to Referee 1, who give us the positive comment.

Referee 2:

Comment:

The authors have revised their manuscript according to the input of reviewers. Now the comparison with Neutron scattering is made in a more appropriate way. As I stated in my previous report, many of the authors of this work are known experts in the field and this is a very solid study worthy publishing. However, with the delay of re-submission to another journal, the novelty of this work is further compromised. While I was re-evaluating this work for Nature Communications, it has come to my attention that a very similar study on the temperature dependence has been published in Physical Review Letters last month: PRL 116, 186601 (2016). The very similar conclusion with the present manuscript was drawn in the PRL paper. Therefore I couldn't recommend publication of this work in Nature Communications.

Ans: We thank the pertinent comments by referee 2. We have cited the paper (PRL 116, 186601(2016)) in the new version.

Referee 2:

Comment:

The authors have improved the manuscript by incorporating a connection to the possible mechanisms (which were asserted before and not explicitly explained).

Although the paper remains primarily experimental with many remaining questions, its results and the connection of the AFM fluctuations to the enhancement of the ISHE is well established by the data and the systematic methodology of the experiments (including the Cu inserted sample).

I am satisfied with the reply to my previous comments and I am happy to recommend the article for publication in nature Communications.

Ans: We thank Referee 3, who gives us the positive comment again.